# Consistency-Guided Asynchronous Contrastive Tuning for Few-Shot Class-Incremental Tuning of Foundation Models

**Shuvendu Roy**[*,1,2], **Elham Dolatabadi**[1,3], **Arash Afkanpour**[1], **Ali Etemad**[2]
[1]**Vector Institute,**     [2]**Queen's University, Canada,**     [3]**York University, Canada**

**Reviewed on OpenReview:** `https://openreview.net/forum?id=WfAvMdwiE8`

## Abstract

We propose Consistency-guided Asynchronous Contrastive Tuning (CoACT), a novel method for continuously tuning foundation models to learn new classes in few-shot settings. CoACT consists of three key components:(*i*) asynchronous contrastive tuning, which learns new classes by including LoRA modules in the pre-trained encoder while enforcing consistency between two asynchronous encoders; (*ii*) controlled fine-tuning, which facilitates effective tuning of a subset of the foundation model; and (*iii*) consistency-guided incremental tuning, which enforces additional regularization during later sessions to reduce forgetting of the learned classes. We evaluate our proposed solution on Few-Shot Class-Incremental Learning (FSCIL) as well as a new and more challenging setup called Few-Shot Class-Incremental Tuning (FSCIT), which facilitates the continual tuning of vision foundation models to learn new classes with only a few samples per class. Unlike traditional FSCIL, FSCIT does not require a large in-distribution base session for initial fully supervised training prior to the incremental few-shot sessions. We conduct extensive evaluations across 16 diverse datasets, demonstrating the effectiveness of CoACT in both FSCIL and FSCIT setups. CoACT outperforms existing methods by up to 5.02% in FSCIL and up to 12.51% in FSCIT for individual datasets, with an average improvement of 2.47%. Furthermore, CoACT exhibits reduced forgetting and enhanced robustness in low-shot experiments. Detailed ablation and sensitivity studies highlight the contribution of each component of CoACT. We make our code publicly available at https://github.com/ShuvenduRoy/CoACT-FSCIL.

## 1 Introduction

Large foundation models pre-trained on web-scale unlabeled data exhibit strong generalization capabilities on downstream tasks when fine-tuned with relatively small amounts of labelled data (Radford et al., 2021; Zhou et al., 2022c). However, the immense size of these pre-trained models introduces significant challenges for fine-tuning, especially when working with limited labelled data (Khattak et al., 2023; Yao et al., 2023). Despite recent advancements such as parameter-efficient tuning (Khattak et al., 2023; Gao et al., 2023), it has been observed that fine-tuning the model in a few-shot setting often leads to a decline in the out-of-the-box generalization capability of the foundation model (Roy & Etemad, 2024). Yet, real-world applications not only necessitate learning from a few samples but also demand continual learning of new tasks. Few-shot class-incremental learning (FSCIL) (Masana et al., 2022) is a continual learning setup that extends the scope of few-shot learning by enabling models to adapt incrementally to new tasks while preserving existing knowledge. While few prior works (D'Alessandro et al., 2023; Park et al., 2024) have explored tuning a foundation model in FSCIL setups, existing approaches often struggle with the forgetting of leaned classes and a loss of the foundation model's generalization capabilities.

In this work, we propose Consistency-guided Asynchronous Contrastive Tuning (CoACT), a novel framework for class-incremental tuning of vision foundation models in few-shot settings. CoACT introduces three novel

---

*Corrosponding Author. Email: shuvendu.roy@queensu.ca

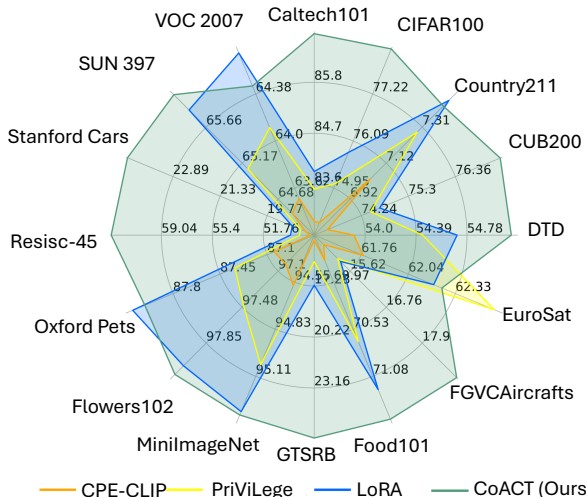

Figure 1: Performance comparison on FSCIT with a foundation model.

Table 1: Performance comparison to existing methods on tradition FSCIL.

| Method | CIFAR-100 | CUB-200 | miniIN |
|---|---|---|---|
| LIMIT | 51.23 | 57.41 | 49.19 |
| FACT | 52.10 | 56.94 | 50.49 |
| SAVC | 53.12 | 62.50 | 57.11 |
| SoftNet | 55.33 | 56.75 | 54.68 |
| BOT | 58.75 | 63.75 | 59.57 |
| SV-T | 69.75 | 76.17 | 82.38 |
| CPE-CLIP | 80.52 | 64.60 | 82.77 |
| PriViLege | 86.06 | 75.08 | 94.10 |
| **CoACT (Ours)** | 84.63 | 81.19 | 96.24 |

components: (**i**) asynchronous contrastive tuning, (**ii**) controlled fine-tuning, and (**iii**) consistency-guided incremental tuning. To strike a balance between adaptability to learn new classes and retaining generalizable knowledge of the pre-trained foundation model **asynchronous contrastive tuning** learns from the first incremental session using a novel asynchronous contrastive loss. Specifically, to provide adaptability to learn new classes, we integrate learnable LoRA modules into the pre-trained encoder and ensure generalization by enforcing consistency between two asynchronous encoders: *a student encoder* containing the learnable modules and *a teacher encoder* identical to the pre-trained encoder, updated as the Exponential Moving Average (EMA) of the student. This prevents rapid change in the output distribution of the teacher, which in turn helps reduce overfitting in the student encoder while learning new classes. To further enhance adaptability, we introduce **controlled fine-tuning**, which is a two-step training protocol for training the first incremental session. First, we train the newly added LoRA modules with a high Learning Rate (LR) for a certain number of epochs and then fine-tune the last few layers of the pre-trained parameters with a relatively lower LR. This also helps balance adaptability with generalizability. Finally, to ensure effective learning of classes in the following incremental sessions while preventing forgetting of previously learned classes and preserving the generalization capabilities of the foundation model at the same time, we introduce a novel regularization technique, **consistency-guided incremental tuning**. We achieve this by enforcing consistency between the predictions of the learnable encoder in the incremental sessions and the frozen encoder from the first incremental session.

We conduct a comprehensive study on 16 diverse image recognition datasets to investigate the effectiveness of our method in FSCIL. Additionally, we introduce a novel and more challenging setup called few-shot class-incremental tuning (FSCIT), where, unlike FSCIL, we do not assume the availability of a fully-supervised base session. Specifically, in FSCIT, a foundation model is fine-tuned over incremental sessions with a few samples per class for all sessions, including the first session. The datasets include generic objects, fine-grained objects, scenes, satellite images, and texture recognition. Our comprehensive experiments demonstrate that on the FSCIL setup, CoACT outperforms prior methods by up to 5.02%. On our proposed FSCIT setup, CoACT outperforms existing methods by 2.47% on average across 16 datasets, with up to 12.79% performance gain on individual datasets. More importantly, CoACT exhibits reduced forgetting of already learned classes as the number of classes increases. CoACT also exhibit generalization with different sizes of pre-trained encoders and different numbers of shots. We provide detailed ablation studies showing the effectiveness of each component of our method. Overall, our contributions are:

- We propose CoACT, which consists of three novel components to tune a vision foundation model in a few-shot continual learning setup without losing its generalization capability or forgetting the already learned classes.
- We introduce a new paradigm of continual learning called Few-shot Class-Incremental Tuning (FSCIT), which aims to tune a foundation model to continuously learn new classes with few samples per class. We also establish several baselines for this new paradigm.
- Comprehensive experiments show the effectiveness of our method, achieving state-of-the-art (SOTA) on both FSCIT and FSCIL setup. We also show reduced forgetting and effectiveness in very-low-shot settings. Extensive ablation and sensitivity studies show the effectiveness of each of our proposed components.

## 2 Related works

### 2.1 Fine-tuning foundation models.

A number of techniques have recently been proposed to tune foundation models without the need to re-train them from scratch, addressing the growing demand for computationally efficient and parameter-efficient fine-tuning methods. These approaches leverage the pre-trained capabilities of large models, adapting them to downstream tasks without modifying the core pre-trained parameters extensively. Adapter tuning (Houlsby et al., 2019) fine-tunes large pre-trained models to downstream tasks by inserting new learnable layers inside the pre-trained model. On the other hand, prompt-tuning (Lester et al., 2021) and prefix-tuning (Li & Liang, 2021) add learnable prompts with the input embedding for learning the new task without tuning the pre-trained parameters of the model. Low-rank adapters (Hu et al., 2021; Karimi Mahabadi et al., 2021) have been introduced to reduce the computation cost of additional parameters. VPT (Jia et al., 2022) and AdapterFormer (Chen et al., 2022) explored parameter-efficient fine-tuning in the context of vision transformers. A well-known issue of training a foundation model with few labelled samples per class is the overfitting and the loss of the generalization of the foundation model. More recent, CoPrompt Roy & Etemad (2024) proposed a novel regularization technique by enforcing consistency between the pre-trained and learnable encoders to reduce overfitting and improve generalization. Nonetheless, existing methods for tuning foundation models are not designed for *continuous* tuning of the model since there are no inherent mechanisms to prevent loss of generalization and catastrophic forgetting.

### 2.2 Few-shot class-incremental learning.

Class-incremental learning is a continual learning process that focuses on continuous learning of new classes while retaining the knowledge of already learned ones (Wang et al., 2023; Smith et al., 2023; Wang et al., 2024; 2022). In practice, machine learning models often need to learn new classes from a few labelled samples per class (Zhou et al., 2022b), while having no access to samples from already learned classes. This scenario has given rise to a new learning task called few-shot class-incremental learning or FSCIL (Tian et al., 2023; Tao et al., 2020). The existing literature on FSCIL can be broadly categorized into two main groups: methods that continuously train both the encoder and classifier over each incremental session (Cheraghian et al., 2021; Dong et al., 2021; Zhao et al., 2021), and methods that keep the encoder frozen during the incremental learning sessions (Zhu et al., 2021; Shi et al., 2021; Zhang et al., 2021).

As an example of the first group of methods, MgSvF (Zhao et al., 2021) employed a component-wise update strategy to ensure adapting to new classes while preserving knowledge of existing ones. The exemplar relation distillation framework (Dong et al., 2021) constructed and updated a graph of exemplar relationships to facilitate the integration of new classes. SoftNet (Kang et al., 2023) introduced a novel approach for identifying and freezing a sub-network of crucial parameters from the previous session and training the remaining parameters during incremental sessions. While such methods in the first group generally offer greater *adaptability*, the methods in the second group focus on ensuring *stability* by maintaining separability in the base classes in the learned embedding space. For example, FACT (Zhou et al., 2022a) introduced virtual prototypes to maximize class embedding separation while preserving their relative positions. SAVC (Song et al., 2023) generated virtual classes during the base session training to maximize separability. Similarly,

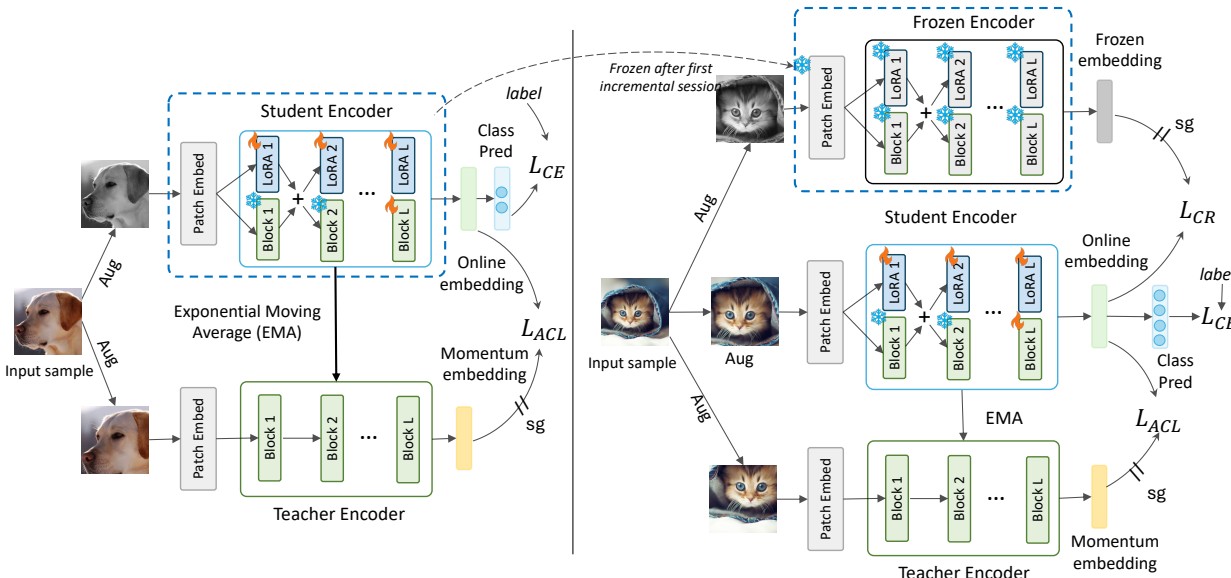

Figure 2: Illustration of CoACT. (Left) Training on the first incremental session with asynchronous contrastive tuning and controlled fine-tuning. The student encoder contains learnable LoRA modules, while the teacher is identical to the foundation model but updated as the EMA of the student. Controlled fine-tuning enables the tuning of a subset of the foundation model with reduced LR after certain epochs. (Right) Consistency-guided incremental tuning enforces consistency between the learnable student and the frozen encoder from the first session, providing additional regularization that prevents overfitting and forgetting.

NC-FSCIL (Yang et al., 2022) pre-assigned optimally spaced prototypes to each base class, promoting diverse and distinct class representations.

Some of the more recent works have explored tuning a foundation in the FSCIL setup. For instance, CPE-CLIP (D'Alessandro et al., 2023) utilized the strong generalization capability of a pre-trained vision-language model to learn new classes with parameter-efficient fine-tuning. SV-T (Qiu et al., 2023) used a semantic-visual guided Transformer to enable few-shot class-incremental learning while reducing overfitting by combining text-based semantic labels with visual labels to improve generalization to new classes. PriViLege (Park et al., 2024) showed that with prompt tuning and knowledge distillation, pre-trained vision language shows strong performances in FSCIL. Nonetheless, these methods operate on the FSCIL setting and still show forgetting issues as they learn new classes.

## 3 Method

### 3.1 Problem formulation

In FSCIT, a model $\phi(x)$ is continuously trained over $T$ consecutive sessions of new classes. Each session follows an $n$-way, $k$-shot setup, where $n$ represents the number of classes in each session, and $k$ is the number of samples per class. Consequently, the training data for each session $t \in T$ can be represented as $\mathcal{D}_{train}^t = \{(x_i, \ y_i)\}_{i=1}^{N_t}$, where $x_i$ and $y_i$ denote the $i^{\text{th}}$ sample and its corresponding label, and $N_t$ represents the number of samples in session $t$. We can express $\phi(x) = W^T f_\theta(x)$, where $f_\theta$ is an encoder and $W$ is a linear classifier. In the conventional setup of FSCIL, $f_\theta$ is trained on a large labelled in-distribution base session $(\mathcal{D}_{train}^0)$ in a fully-supervised manner. In FSCIT, we do not assume the availability of such a base session, rendering standard FSCIL techniques incompatible. In this work, we aim to design a method for tuning off-the-shelf vision foundation models to continuously learn new classes in few-shot settings.

### 3.2 Consistency-guided Asynchronous Contrastive Tuning (CoACT)

In this section, we discuss the details of the three components of our proposed method: asynchronous contrastive tuning, controlled fine-tuning, and consistency-guided incremental tuning. Here, asynchronous contrastive tuning and controlled fine-tuning facilitate learning the first session (see Figure 2 (left)), while consistency-guided incremental tuning learns the remaining incremental sessions without forgetting the learned classes (see Figure 2 (right)).

#### 3.2.1 Asynchronous contrastive tuning.

To strike a balance between adaptability to learn new classes and retaining generalizable knowledge of the foundation model, we introduce asynchronous contrastive tuning as the first component in our framework. This involves fine-tuning the pre-trained model using our novel Asynchronous Contrastive Learning (ACL) approach while incorporating LoRA modules into the model. Let $h_i = f_\theta^{(i)}(h_{i-1})$ be the output of $i^{\text{th}}$ layer of the pre-trained encoder, $h_{i-1}$ be the output of the $(i-1)^{\text{th}}$ hidden layer of the encoder, and $f_\theta^{(i)}$ be the $i^{\text{th}}$ layer of the encoder. With the new learnable LoRA layers, the output of the $i^{\text{th}}$ layer of the network can be represented as $h_i' = f_\theta^{(i)}(h_{i-1}) + f_{\text{LoRA}}^{(i)}(h_{i-1})$, where, $f_{\text{LoRA}}^{(i)}$ is the $i^{\text{th}}$ LoRA layer added to the pre-trained encoder. For brevity, we denote the encoder with learnable LoRA layers as $f_{\theta'}$. We can train $f_{\theta'}$ on $\mathcal{D}_{train}^1$ to learn the first session:

$$\mathcal{L}_{sup} = \mathcal{L}_{ce}(W^T f_{\theta'}(x), y). \tag{1}$$

However, it has been shown in prior work that cross-entropy alone does not learn a well-separable embedding space (Song et al., 2023) and has a higher over-fitting tendency, especially in a few-shot setting (Roy & Etemad, 2024). To reduce the possibility of overfitting and retain the generalization in the learnable encoder $f_{\theta'}$, we regularize its output distribution with an **asynchronous** teacher encoder by maximizing their agreement in the embedding space. Our novelty lies in the asynchronous design of the encoders, where the student encoder contains the learnable LoRA modules, but the teacher encoder is identical to the pre-trained model (without LoRA modules) and learned through the Exponential Moving Average (EMA) of the student $f_{\theta'}$ as $\theta'' = m \cdot \theta'' + (1-m) \cdot \theta$, where $\theta$ and $\theta''$ are the parameters of the student (excluding LoRA) and teacher encoders, and $m$ is the momentum parameter. Since the teacher lacks LoRA modules, the EMA update applies only to the student's base parameters, affecting the teacher layers corresponding to those fine-tuned in the student (Section 3.2.2). Given that the teacher and student encoders differ in their architecture due to the addition of LoRA to the student, learning occurs asynchronously. This asynchronous encoder design and slow-moving update of the teacher through EMA ensures that the predictions from the teacher do not fluctuate. Since the teacher encoder is also initialized from the foundation model, consistency with the teacher effectively regularizes the student from overfitting.

In practice, we maximize the agreement between the embeddings of the student and teacher encoders on all the samples from each class as:

$$\mathcal{L}_{ACL} = -\sum_i \frac{1}{|C_i|} \sum_{j \in C_i} \log \frac{\exp(\langle q_i, k_j \rangle / \tau)}{\sum_{l \neq i} \exp(\langle q_i, k_l \rangle / \tau)}, \tag{2}$$

where $C_i \overset{\text{def}}{=} \{j : y_j = y_i\}$, $\langle \cdot, \cdot \rangle$ denotes inner product, $q_i = f_{\theta'}(\mathcal{A}_1(x_i))$ and $k_j = f_{\theta''}(\mathcal{A}_2(x_j))$ are online embeddings and momentum embeddings of augmentations of $x_i$ and $x_j$ from the student and the teacher encoder respectively, and $\mathcal{A}_1$ and $\mathcal{A}_2$ are random augmentations. Finally, we train the model with the $\mathcal{L}_{ACL}$ and $\mathcal{L}_{sup}$ as: $\mathcal{L}_{ACL} + \lambda \cdot \mathcal{L}_{sup}$, where $\lambda$ controls the impact of $\mathcal{L}_{sup}$.

The exclusion of LoRA adapters from the EMA updates prevents rapid fluctuations in the teacher encoder, ensuring it serves as a slow-moving reference that mitigates overfitting and retains the generalization capabilities of the pre-trained foundation model. If LoRA adapters were included, the teacher would change more dynamically, reducing its ability to regularize the student effectively and undermining the balance between adaptability and generalization retention. Maintaining the teacher closer to the pre-trained model prevents the student from drifting too far and acts as a form of regularization that constrains learned representations while preserving prior knowledge.

### 3.2.2 Controlled fine-tuning.

Controlled fine-tuning is a two-step training protocol to enhance the adaptability of the model by selectively fine-tuning a few layers of the pre-trained encoder. Since the newly added LoRA modules are randomly initialized, we begin by training only the LoRA modules of the student encoder with a higher learning rate for the initial $E_c$ epochs of training to prevent the propagation of randomness to the well-trained pre-trained encoder, which could otherwise lead to forgetting of its generalizable knowledge. This is followed by a fine-tuning stage where the last $C_l$ layers of the pre-trained encoder are fine-tuned with a reduced LR (scaled by a factor of $C_f$). We focus on fine-tuning only the last $C_l$ layers, as the later layers of a pre-trained model are responsible for learning domain-specific fine-grained features, whereas the earlier layers are more general and transferable to a wide range of tasks (Neyshabur et al., 2020). Importantly, this fine-tuning is performed under the consistency constraint of asynchronous contrastive learning, ensuring the model learns target-domain features effectively without overfitting or forgetting pre-trained knowledge. This balance between adaptability and generalization enables effective adaptation to new classes while preserving the pre-trained encoder's foundational capabilities. The two-setup controlled fine-tuning strategy is applied during the first session, while in later sessions, fine-tuning continues with LoRA across all layers, along with fine-tuning the last $C_l$ layers.

### 3.2.3 Consistency-guided incremental tuning.

While the first two modules facilitate tuning the foundation model ($f_{\theta'}$) during the first session, later sessions also require the retention of previously learned classes. To facilitate this, we propose consistency-guided incremental tuning, which prevents forgetting by regularizing the output distribution of the student $f_{\theta'}$ when training on the incremental sessions. More specifically, we enforce consistency between the predictions of the student encoder and the frozen student encoder after the first session, effectively discouraging substantial changes in the learned representations of the student. We do not enforce consistency with the frozen encoder from the most recent session (e.g., using the frozen encoder from session $t-1$ for session $t$), since the model may gradually drift away from the pre-trained encoder's knowledge with cumulative overfitting over the sessions, which can lead to increased forgetting of the foundational generalization capabilities of the model. By anchoring the consistency to the first session's encoder, CoACT effectively balances the learning of new classes with the retention of pre-trained knowledge, mitigating catastrophic forgetting. In a standard class-incremental learning setup, the encoder updates across multiple sessions as $f_{\theta_t} = f_{\theta^{t-1}} + \Delta_{\theta_t}$ where $\Delta_{\theta_t}$ represents the parameter updates due to learning new classes at session $t$. If is consistency enforced with the encoder from the previous session, $t-1$. As a result, the model incrementally adapts but can also gradually drift away from its initial state, leading to $f_{\theta_T} = f_{\theta^0} + \sum_{t=1}^{T} \Delta_{\theta_t}$. This, in turn, can result in a gradual loss of generalization, where the model drifts away from the foundational encoder. Instead, enforcing consistency with the frozen encoder from the first session $f_{\theta^0}$ prevents progressive divergence. This ensures that at each session, $f_{\theta_t} = f_{\theta^0} + \Delta'_{\theta_t}$, where $\|\Delta'_{\theta_t}\| \ll \|\sum_{i=1}^{t} \Delta_{\theta_i}\|$.

Let $f_{\theta_\beta}$ be the frozen encoder after the first session, and the frozen embedding of this encoder be $p_i = f_{\theta_\beta}(\mathcal{A}(x_i))$. We define our consistency regularizer as:

$$\mathcal{L}_{CR} = -\sum_i \frac{1}{|C_i|} \sum_{j \in C_i} \log \frac{\exp(\langle q_i, p_j \rangle / \tau)}{\sum_{l \neq i} \exp(\langle q_i, p_l \rangle / \tau)}. \tag{3}$$

Finally, we train the model after the first session with: $\mathcal{L}_{CR} + \gamma \mathcal{L}_{ACL} + \lambda \mathcal{L}_{sup}$, where $\gamma$ and $\lambda$ controls the relative importance of the loss functions. We tune the LoRA modules and the classifier during incremental training while keeping the encoder frozen.

## 4 Experiments

### 4.1 Datasets and implementation details.

Following existing literature on FSCIL, we evaluate CoACT on CIFAR-100 (Krizhevsky et al., 2009), CUB-200 (Wah et al., 2011), and miniImageNet (Russakovsky et al., 2015) datasets. We evaluate our new benchmark,

Table 2: Comparison to prior works across the base and incremental sessions on CUB-200 with 100 base classes and 10-way 5-shot evaluation setup.

| Method | PT. Backbone | Acc. in each session (%) ↑ | | | | | | | | | | |
|---|---|---|---|---|---|---|---|---|---|---|---|---|
| | | 0 | 1 | 2 | 3 | 4 | 5 | 6 | 7 | 8 | 9 | 10 |
| iCaRL (Rebuffi et al., 2017) | ✗ | 68.68 | 52.65 | 48.61 | 44.16 | 36.62 | 29.52 | 27.83 | 26.26 | 24.01 | 23.89 | 21.16 |
| EEIL (Castro et al., 2018) | ✗ | 68.68 | 53.63 | 47.91 | 44.20 | 36.30 | 27.46 | 25.93 | 24.70 | 23.95 | 24.13 | 22.11 |
| TOPIC (Tao et al., 2020) | ✗ | 68.68 | 62.49 | 54.81 | 49.99 | 45.25 | 41.40 | 38.35 | 35.36 | 32.22 | 28.31 | 26.28 |
| Rebalancing (Hou et al., 2019) | ✗ | 68.68 | 57.12 | 44.21 | 28.78 | 26.71 | 25.66 | 24.62 | 21.52 | 20.12 | 20.06 | 19.87 |
| SPPR (Zhu et al., 2021) | ✗ | 68.68 | 61.85 | 57.43 | 52.68 | 50.19 | 46.88 | 44.65 | 43.07 | 40.17 | 39.63 | 37.33 |
| MetaFSCIL(Chi et al., 2022) | ✗ | 75.90 | 72.41 | 68.78 | 64.78 | 62.96 | 59.99 | 58.30 | 56.85 | 54.78 | 53.82 | 52.64 |
| F2M (Shi et al., 2021) | ✗ | 81.07 | 78.16 | 75.57 | 72.89 | 70.86 | 68.17 | 67.01 | 65.26 | 63.36 | 61.76 | 60.26 |
| CEC (Zhang et al., 2021) | ✗ | 75.85 | 71.94 | 68.50 | 63.50 | 62.43 | 58.27 | 57.73 | 55.81 | 54.83 | 53.52 | 52.28 |
| FACT (Zhou et al., 2022a) | ✗ | 75.90 | 73.23 | 70.84 | 66.13 | 65.56 | 62.15 | 61.74 | 59.83 | 58.41 | 57.89 | 56.94 |
| LIMIT (Zhou et al., 2022b) | ✗ | 75.89 | 73.55 | 71.99 | 68.14 | 67.42 | 63.61 | 62.40 | 61.35 | 59.91 | 58.66 | 57.41 |
| SoftNet (Kang et al., 2023) | ✗ | 78.07 | 74.58 | 71.37 | 67.54 | 65.37 | 62.60 | 61.07 | 59.37 | 57.53 | 57.21 | 56.75 |
| SAVC (Song et al., 2023) | ✗ | 81.85 | 77.92 | 74.95 | 70.21 | 69.96 | 67.02 | 66.16 | 65.30 | 63.84 | 63.15 | 62.50 |
| BOT (Roy et al., 2024) | ✗ | 82.31 | 78.03 | 75.45 | 70.99 | 71.06 | 67.85 | 67.44 | 66.05 | 64.95 | 64.31 | 63.75 |
| SV-T (Qiu et al., 2023) | SwinT | 84.19 | 82.63 | 81.21 | 78.97 | 79.38 | 77.64 | 77.55 | 75.71 | 75.91 | 75.77 | 76.17 |
| CPE-CLIP (D'Alessandro et al., 2023) | CLIP-B/16 | 81.58 | 78.52 | 76.68 | 71.86 | 71.52 | 70.23 | 67.66 | 66.52 | 65.09 | 64.47 | 64.60 |
| PriViLege (Park et al., 2024) | CLIP-B/16 | 82.21 | 81.25 | 80.45 | 77.76 | 77.78 | 75.95 | 75.69 | 76.00 | 75.19 | 75.19 | 75.08 |
| **Ours** | ViT-B/16 | 88.68 | 86.26 | 85.83 | 83.38 | 83.52 | 81.71 | 81.77 | 81.77 | 81.02 | 80.76 | 81.19 |

Table 3: Comparison to prior works across the base and incremental sessions on miniImageNet with 60 base classes and 5-way 5-shot incremental setting.

| Method | PT. Backbone | Acc. in each session (%) ↑ | | | | | | | | |
|---|---|---|---|---|---|---|---|---|---|---|
| | | 0 | 1 | 2 | 3 | 4 | 5 | 6 | 7 | 8 |
| iCaRL (Rebuffi et al., 2017) | ✗ | 71.77 | 61.85 | 58.12 | 54.60 | 51.49 | 48.47 | 45.90 | 44.19 | 42.71 |
| Rebalancing (Hou et al., 2019) | ✗ | 72.30 | 66.37 | 61.00 | 56.93 | 53.31 | 49.93 | 46.47 | 44.13 | 42.19 |
| TOPIC (Tao et al., 2020) | ✗ | 61.31 | 50.09 | 45.17 | 41.16 | 37.48 | 35.52 | 32.19 | 29.46 | 24.42 |
| EEIL (Castro et al., 2018) | ✗ | 61.31 | 46.58 | 44.00 | 37.29 | 33.14 | 27.12 | 24.10 | 21.57 | 19.58 |
| FSLL (Mazumder et al., 2021) | ✗ | 66.48 | 61.75 | 58.16 | 54.16 | 51.10 | 48.53 | 46.54 | 44.20 | 42.28 |
| FSLL+SS (Mazumder et al., 2021) | ✗ | 68.85 | 63.14 | 59.24 | 55.23 | 52.24 | 49.65 | 47.74 | 45.23 | 43.92 |
| F2M (Shi et al., 2021) | ✗ | 72.05 | 67.47 | 63.16 | 59.70 | 56.71 | 53.77 | 51.11 | 49.21 | 47.84 |
| CEC (Zhang et al., 2021) | ✗ | 72.00 | 66.83 | 62.97 | 59.43 | 56.70 | 53.73 | 51.19 | 49.24 | 47.63 |
| MetaFSCIL (Chi et al., 2022) | ✗ | 72.04 | 67.94 | 63.77 | 60.29 | 57.58 | 55.16 | 52.90 | 50.79 | 49.19 |
| C-FSCIL (Hersche et al., 2022) | ✗ | 76.40 | 71.14 | 66.46 | 63.29 | 60.42 | 57.46 | 54.78 | 53.11 | 51.41 |
| FACT (Zhou et al., 2022a) | ✗ | 72.56 | 69.63 | 66.38 | 62.77 | 60.60 | 57.33 | 54.34 | 52.16 | 50.49 |
| CLOM (Zou et al., 2022) | ✗ | 73.08 | 68.09 | 64.16 | 60.41 | 57.41 | 54.29 | 51.54 | 49.37 | 48.00 |
| LIMIT (Zhou et al., 2022b) | ✗ | 72.32 | 68.47 | 64.30 | 60.78 | 57.95 | 55.07 | 52.70 | 50.72 | 49.19 |
| SoftNet (Kang et al., 2023) | ✗ | 79.77 | 75.08 | 70.59 | 66.93 | 64.00 | 61.00 | 57.81 | 55.81 | 54.68 |
| SAVC (Song et al., 2023) | ✗ | 81.12 | 76.14 | 72.43 | 68.92 | 66.48 | 62.95 | 59.92 | 58.39 | 57.11 |
| BOT (Roy et al., 2024) | ✗ | 84.30 | 79.59 | 75.49 | 71.4 | 68.45 | 65.12 | 62.20 | 60.52 | 59.57 |
| SV-T (Qiu et al., 2023) | SwinT | 88.75 | 87.92 | 86.07 | 84.84 | 84.30 | 83.24 | 82.22 | 82.28 | 82.38 |
| CPE-CLIP (D'Alessandro et al., 2023) | CLIP-B/16 | 90.23 | 89.56 | 87.42 | 86.80 | 86.51 | 85.08 | 83.43 | 83.38 | 82.77 |
| PriViLege (Park et al., 2024) | CLIP-B/16 | 96.68 | 96.49 | 95.65 | 95.54 | 95.54 | 94.91 | 94.33 | 94.19 | 94.10 |
| **Ours** | ViT-B/16 | 97.63 | 97.55 | 97.09 | 97.02 | 97.0 | 96.58 | 96.3 | 96.29 | 96.24 |

FSCIT on a diverse set of 16 datasets, including generic object detection (Caltech101 (Fei-Fei et al., 2004), CIFAR-100 (Krizhevsky et al., 2009), CUB-200 (Wah et al., 2011), miniImageNet (Russakovsky et al., 2015), VOC 2007 (Everingham, 2008)), fine-grained recognition (OxfordPets (Parkhi et al., 2012), StanfordCars (Krause et al., 2013), Flower102 (Nilsback & Zisserman, 2008), Food101 (Bossard et al., 2014), FGVCAircraft (Maji et al., 2013)), scene recognition (SUN397 (Xiao et al., 2010), Country211 (Radford et al., 2021)), satellite-image (EuroSAT (Helber et al., 2019), Resisc-45 (Cheng et al., 2017)), texture recognition (DTD

Table 4: Comparison to prior works across the base and incremental sessions on CIFAR-100 with 60 base classes and 5-way 5-shot incremental setting.

| Method | PT. Backbone | Acc. in each session (%) ↑ | | | | | | | | |
|---|---|---|---|---|---|---|---|---|---|---|
| | | **0** | **1** | **2** | **3** | **4** | **5** | **6** | **7** | **8** |
| Rebalancing (Hou et al., 2019) | ✗ | 61.31 | 47.80 | 39.31 | 31.91 | 25.68 | 21.35 | 18.67 | 17.24 | 14.17 |
| iCaRL (Rebuffi et al., 2017) | ✗ | 61.31 | 46.32 | 42.94 | 37.63 | 30.49 | 24.00 | 20.89 | 18.80 | 17.21 |
| TOPIC (Tao et al., 2020) | ✗ | 61.31 | 50.09 | 45.17 | 41.16 | 37.48 | 35.52 | 32.19 | 29.46 | 24.42 |
| IDLVQ-C (Chen & Lee, 2020) | ✗ | 64.77 | 59.87 | 55.93 | 52.62 | 49.88 | 47.55 | 44.83 | 43.14 | 41.84 |
| FSLL (Mazumder et al., 2021) | ✗ | 66.48 | 61.75 | 58.16 | 54.16 | 51.10 | 48.53 | 46.54 | 44.20 | 42.28 |
| FSLL+SS (Mazumder et al., 2021) | ✗ | 68.85 | 63.14 | 59.24 | 55.23 | 52.24 | 49.65 | 47.74 | 45.23 | 43.92 |
| F2M (Shi et al., 2021) | ✗ | 67.28 | 63.80 | 60.38 | 57.06 | 54.08 | 51.39 | 48.82 | 46.58 | 44.65 |
| CEC (Zhang et al., 2021) | ✗ | 73.07 | 68.88 | 65.26 | 61.19 | 58.09 | 55.57 | 53.22 | 51.34 | 49.14 |
| MetaFSCIL (Chi et al., 2022) | ✗ | 74.50 | 70.10 | 66.84 | 62.77 | 59.48 | 56.52 | 54.36 | 52.56 | 49.97 |
| CLOM (Zou et al., 2022) | ✗ | 74.2 | 69.83 | 66.17 | 62.39 | 59.26 | 56.48 | 54.36 | 52.16 | 50.25 |
| C-FSCIL (Hersche et al., 2022) | ✗ | 77.47 | 72.40 | 67.47 | 63.25 | 59.84 | 56.95 | 54.42 | 52.47 | 50.47 |
| LIMIT (Zhou et al., 2022b) | ✗ | 73.81 | 72.09 | 67.87 | 63.89 | 60.70 | 57.77 | 55.67 | 53.52 | 51.23 |
| FACT (Zhou et al., 2022a) | ✗ | 74.60 | 72.09 | 67.56 | 63.52 | 61.38 | 58.36 | 56.28 | 54.24 | 52.10 |
| SAVC (Song et al., 2023) | ✗ | 79.85 | 73.70 | 69.37 | 65.28 | 61.91 | 59.27 | 57.24 | 54.97 | 53.12 |
| SoftNet (Kang et al., 2023) | ✗ | 79.88 | 75.54 | 71.64 | 67.47 | 64.45 | 61.09 | 59.07 | 57.29 | 55.33 |
| BOT (Roy et al., 2024) | ✗ | 80.25 | 77.20 | 75.09 | 70.82 | 67.83 | 64.86 | 62.73 | 60.52 | 58.75 |
| SV-T (Qiu et al., 2023) | SwinT | 86.77 | 82.82 | 80.36 | 77.20 | 76.06 | 74.00 | 72.92 | 71.68 | 69.75 |
| CPE-CLIP (D'Alessandro et al., 2023) | CLIP-B/16 | 87.83 | 85.86 | 84.93 | 82.85 | 82.64 | 82.42 | 82.27 | 81.44 | 80.52 |
| PriViLege (Park et al., 2024) | CLIP-B/16 | 90.88 | 89.39 | 88.97 | 87.55 | 87.83 | 87.35 | 87.53 | 87.15 | 86.06 |
| **Ours** | ViT-B/16 | 90.46 | 88.46 | 88.11 | 86.94 | 86.98 | 86.52 | 86.39 | 86.0 | 84.63 |

(Cimpoi et al., 2014)), and traffic sign recognition (GTSRB (Houben et al., 2013)). By default, we divide the classes into 10 (or 9) sessions with an equal number of classes and perform 10-shot continual learning.

The performance after each session is calculated as the average accuracy of all classes seen so far. Unless specified otherwise, accuracy refers to the accuracy after the last session, which is the average accuracy over all classes on the test set. We also evaluate CoACT on the traditional FSCIL setup, where the initial training is performed with a large in-distribution base session, followed by few-shot tuning over the incremental sessions. We use ViT-B/16 as the backbone for most of the experiments, while we present the detailed per-dataset results with ViT-B/32 and ViT-L/16 as well. The encoders are pre-trained on ImgeNet-21K (Russakovsky et al., 2015). We implement our framework in PyTorch and train the model using an SGD optimizer with a momentum of 0.9. The base learning rate is set to 0.1, with a batch size of 64, and the model is trained for 50 epochs for the first session and 5 epochs for the remaining sessions. A cosine LR decay scheduler is used to reduce the learning rate over the training epochs. The teacher encoder is updated with a momentum value of 0.999. For experiments with the FSCIL setup, we train the model for 25 epochs with a learning rate of 0.001. We train the model with input resolution of $224 \times 224$. All other implementation details are the same as described above. All experiments are conducted with 3 random seeds, and the reported results are averaged over the three runs. All experiments are conducted on an Nvidia V100 GPU, where the training takes about 6 hours.

## 4.2 FSCIL results

In this section, we discuss the results of CoACT in the FSCIL setup. Following the existing literature, we present these results on the CIFAR-100, CUB-200, and miniImageNet datasets. In Table 2, we compare the performance of CoACT with prior works on the CUB-200 dataset, where we divide the existing methods into two groups: the first group, which trains randomly initialized models and the second group which use a pre-trained encoder. The results are reported for 100 base classes and a 10-way, 5-shot incremental learning setup. As we observe, CoACT outperforms the previous SOTA (SV-T) by 5.02%.

Table 5: Details on class splits over the continual sessions for different datasets.

| Dataset | Classes | Sessions | First Ses. class | Cls/session |
|---|---|---|---|---|
| Caltech101 | 102 | 10 | 12 | 10 |
| CIFAR-100 | 100 | 10 | 10 | 10 |
| Country211 | 211 | 10 | 22 | 21 |
| CUB-200 | 200 | 10 | 20 | 20 |
| DTD | 47 | 9 | 5 | 5 |
| EuroSAT | 10 | 10 | 1 | 1 |
| FGVCAircraft | 100 | 10 | 10 | 10 |
| Food101 | 101 | 10 | 11 | 10 |
| GTSRB | 43 | 10 | 7 | 4 |
| MiniImageNet | 100 | 10 | 10 | 10 |
| Flower102 | 102 | 10 | 12 | 10 |
| OxfordPets | 37 | 9 | 5 | 4 |
| Resisc-45 | 45 | 9 | 5 | 5 |
| StanfordCars | 196 | 9 | 28 | 21 |
| SUN397 | 397 | 10 | 45 | 44 |
| VOC 2007 | 20 | 10 | 2 | 2 |

Table 6: Performance of CoACT on FSCIT across 16 datasets and its comparison to existing FSCIL methods and our baseline methods. Pro. lear. and Lin. tun. respectively refer to prototype learning and linear tuning.

| Method | Caltech101 | CIFAR100 | Country211 | CUB200 | DTD | EuroSat | FGVCAircraft | Food101 | GTSRB | MiniImageNet | Flowers102 | OxfordPets | Resisc-45 | StanfordCars | SUN397 | VOC 2007 | Average |
|---|---|---|---|---|---|---|---|---|---|---|---|---|---|---|---|---|---|
| CPE-CLIP | 82.79 | 74.11 | 7.03 | 73.47 | 53.92 | 61.77 | 14.79 | 69.71 | 14.64 | 94.57 | 97.02 | 87.05 | 48.43 | 18.51 | 64.48 | 63.53 | 57.85 |
| PriViLege | 83.49 | 75.06 | 7.28 | 74.45 | 54.44 | 62.57 | 15.23 | 70.67 | 15.87 | 95.04 | 97.46 | 87.34 | 49.17 | 18.67 | 65.09 | 64.11 | 58.49 |
| Pro. lear. | 83.47 | 75.58 | 7.42 | 74.10 | 54.72 | 62.20 | 15.51 | 71.09 | 15.53 | 95.31 | 98.09 | 88.10 | 49.83 | 18.90 | 65.68 | 64.72 | 58.77 |
| Lin. tun. | 83.60 | 75.58 | 7.45 | 74.41 | 54.72 | 62.20 | 15.32 | 71.18 | 16.92 | 95.32 | 98.09 | 88.10 | 49.83 | 18.96 | 65.87 | 64.72 | 58.89 |
| LoRA | 83.87 | 75.81 | 7.45 | 74.65 | 54.70 | 62.20 | 15.32 | 71.24 | 17.23 | 95.32 | 98.09 | 88.10 | 49.82 | 18.94 | 65.90 | 64.72 | 58.96 |
| CoACT | 86.86 | 78.31 | 7.42 | 77.38 | 55.11 | 62.25 | 18.98 | 71.59 | 26.05 | 95.34 | 98.18 | 87.79 | 62.62 | 24.40 | 66.11 | 64.45 | 61.43 |

Results on miniImageNet are reported in Table 3, with 60 base classes and a 5-way, 5-shot incremental learning setup. As we find from this table, CoACT outperforms the prior SOTA (PriViLege) by 2.14%. Also, the performance difference increases (over the sessions) as we learn new classes, indicating that CoACT shows less forgetting than PriViLege. Finally, the results for CIFAR-100 are presented in Table 4, where PriViLege (Park et al., 2024) holds the highest result. With significantly fewer parameters (86M in ViT-B/16 compared to 149M in CLIP-B/16), our proposed CoACT achieves competitive results, performing only 1.43% below PriViLege.

## 4.3 FSCIT results

As the first work on FSCIT, we first establish a few baselines to better evaluate our proposed framework. To this end, we explore two of the existing SOTA from the FSCIL literature PriViLege and CPE-CLIP and also explore 3 well-known approaches from the few-shot tuning literature. First, we adopt **prototype learning**, which has been shown to perform well for learning incremental classes in traditional FSCIL (Song et al., 2023). To apply this approach to FSCIT, we keep the pre-trained encoder frozen during the incremental learning sessions, while the classifier ($W$) is learned with the prototypes (Song et al., 2023) of the new classes: $W = \{w_0^0, w_1^0, \cdots, w_{|c^0|}^0\} \cup \{w_0^T, w_1^T, \cdots, w_{|c^T|}^T\}$. Here, the prototype $w_c^t$ of class $c$ from session $t$ is defined as

Table 7: Performance of CoACT on FSCIT across 16 datasets on different encoders.

| Encoder | Caltech101 | CIFAR100 | Country211 | CUB200 | DTD | EuroSat | FGVCAircraft | Food101 | GTSRB | MiniImageNet | Flowers102 | OxfordPets | Resisc-45 | StanfordCars | SUN397 | VOC 2007 | Average |
|---|---|---|---|---|---|---|---|---|---|---|---|---|---|---|---|---|---|
| ViT-B/16 | 86.86 | 78.31 | 7.42 | 77.38 | 55.11 | 62.25 | 18.98 | 71.59 | 26.05 | 95.34 | 98.18 | 87.79 | 62.62 | 24.40 | 66.11 | 64.45 | 61.43 |
| ViT-B/32 | 85.59 | 77.26 | 6.85 | 72.21 | 54.78 | 67.71 | 14.56 | 68.41 | 29.99 | 94.81 | 98.01 | 86.21 | 55.34 | 20.18 | 64.49 | 66.71 | 60.19 |
| ViT-L/14 | 87.25 | 79.27 | 7.45 | 78.29 | 56.19 | 66.36 | 21.24 | 74.99 | 17.22 | 97.18 | 98.18 | 87.95 | 62.25 | 25.43 | 67.18 | 62.23 | 61.79 |
| Swin-B | 88.01 | 80.17 | 8.15 | 78.91 | 57.25 | 68.98 | 23.44 | 76.11 | 28.18 | 97.01 | 98.03 | 88.56 | 64.24 | 25.90 | 69.34 | 66.95 | 63.70 |
| DINO ViT-B/14 | 87.87 | 79.92 | 7.89 | 79.01 | 57.81 | 68.57 | 22.58 | 76.47 | 30.38 | 96.94 | 98.05 | 88.29 | 63.59 | 25.87 | 69.04 | 65.47 | 63.61 |
| CLIP ViT-B/16 (V) | 84.23 | 78.14 | 6.49 | 72.89 | 55.23 | 68.48 | 24.93 | 72.49 | 27.38 | 95.18 | 97.23 | 86.66 | 57.23 | 23.47 | 64.29 | 62.38 | 61.04 |

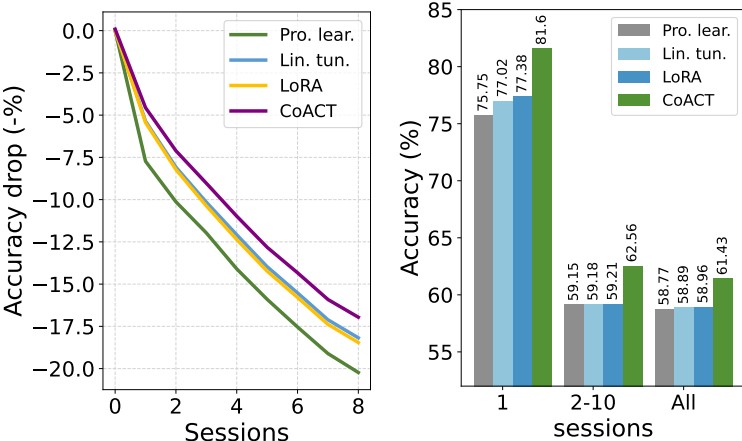

Figure 3: (left) Forgetting of learned classes in FSCIT. (right) Accuracy breakdown into the first, remaining and all sessions in FSCIT setup.

the average of the embeddings of all samples of the class: $w_c^t = \frac{1}{|P_i|} \sum_{j \in P_i} f_\theta(x_j)$, where $P_i \overset{\text{def}}{=} \{j : y_j = i\}$. Next, we consider an incremental-frozen approach with **linear tuning** as a baseline, where we fine-tune the model only on the first session, followed by prototype learning for the remaining sessions (Zhou et al., 2022a). Finally, we use parameter-efficient fine-tuning as the third baseline, where we add LoRA to different layers of the pre-trained encoder, train them during the first setting, and perform prototype learning in incremental sessions. Additionally, to ensure extensive evaluation of the new benchmark, we conduct the experiments on 16 datasets. The classes are divided into sessions as summarized in Table 5.

In Table 6, we present the main results of our study on the FSCIT setup using the ViT-B/16 encoder. The table highlights that existing methods for FSCIL, such as CPE-CLIP and PriViLege, do not perform well on FSCIT. This is primarily because these methods heavily rely on supervised training of the base session, which is absent in the FSCIT setup, leading to reduced performance. Among the other three baselines, namely prototype tuning, linear tuning, and LoRA, performance is slightly better compared to FSCIL methods. However, CoACT surpasses all prior approaches by significant margins on all the datasets. Specifically, CoACT achieves an average improvement of 2.47% over the baselines and exhibits notable gains of up to 12.79% on individual datasets, such as Resisc-45. Additionally, we observe larger improvements on more challenging datasets. For instance, on the five datasets with the lowest accuracy (Country211, FGVCAircraft, GTSRB, Resisc-45, and StanfordCars), CoACT achieves an average improvement of 6.14% over the best-performing baseline.

Next, we report the forgetting of learned classes for each method, measured as the drop in accuracy w.r.t. the first session, in Figure 3 (left). As we observe in this figure, CoACT has the least amount of forgetting

compared to the baselines, with approximately 1.5% less forgetting than both LoRA and linear tuning and 3.2% than prototype learning. We also present a breakdown of accuracies into the first incremental session, the remaining incremental sessions, and all sessions in Figure 3 (right). Here, for all methods, the first session shows particularly higher accuracy than other sessions since there is no interference (or forgetting) of other classes. While LoRA and linear tuning have higher overall accuracy than prototype tuning, the improvement mainly comes from the higher accuracy in the first session only. All baselines, however, perform relatively similarly in the remaining sessions. In contrast, CoACT shows higher improvement in both the first and remaining sessions.

Next, we investigate the generalization of CoACT across different encoders of different sizes. Specifically, we explore ViT-B/16, ViT-B/32, ViT-L/14, Swin-B (Liu et al., 2021), DINO ViT-B/14 (Oquab et al.), and vision encoder from CLIP ViT-B/16 (Radford et al., 2021) as the backbone. The results of this study are presented in Table 7. The results from this study show that our proposed solution generalizes across the encoder sizes. Specifically, with a larger encoder (ViT-L/14), CoACT improves the average accuracy to 61.79%. On individual datasets, we find up to 3.4% improvements over the default ViT-B/16 encoder. Similarly, CoACT shows strong performances on the Swin-B, DINO ViT-B/14, and CLIP ViT-B/16 encoders.

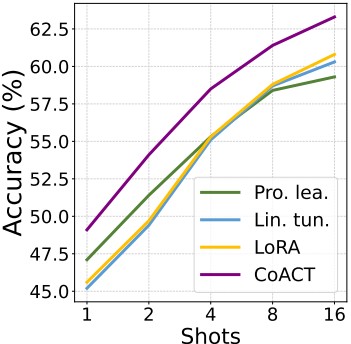

Figure 4: Performance for different shots in FSCIT.

To further evaluate the efficacy of CoACT, we investigate the performance with a different number of samples per class. Specifically, we explore fine-tuning the foundation model in the FSCIT setup with only 1, 2, 4, 8, and 16 samples per class. The results of this experiment are presented in Figure 4. As we find from this experiment, CoACT outperforms other methods in all settings and shows robustness in very low-shot settings. The baselines vary in effectiveness with the different number of samples per class. For instance, prototype learning performs better than linear tuning and LoRA in 1-shot and 2-shot settings, while the other two perform better as the number of samples increases. These results indicate that linear tuning and LoRA do not learn very effective representations in very low-shot settings compared to our method and prototype learning.

## 4.4 Ablation study

We present an ablation study on the proposed components of CoACT in Table 8. Given that the asynchronous contrastive tuning component of our method could not be removed as it contains the trainable parameters, we start this study by removing controlled fine-tuning and consistency-guided incremental tuning modules individually and simultaneously. Interestingly, we observe that while individual removal of these components does not show considerable drops in performance, their concurrent application within our framework results in a significant boost in performance of 1.17% across 16 datasets. Finally, with the ablation of all three components and only training a linear classifier, we observe a 2.66% drop in performance. Notably, all the ablations experiments show a very small standard deviation.

Next, we present a comprehensive study on different parameters of CoACT in Table 9. In the first study (Table 9(a)), we study different alternates for LoRA, namely Adapter (Houlsby et al., 2019) and Prompt (Jia et al., 2022), where we observe that Adapter and Prompt do not perform as well as LoRA, with a final accuracy of 60.01% and 59.87% versus 60.15%, respectively. In this table, the prompt implementation follows the visual prompt tuning (VPT) (Jia et al., 2022) approach, where learnable prompts are added to the input embeddings of the vision transformer. In the next table (Table 9(b)), we study the performance for different numbers of training epochs, where we find the best results when trained for 50 epochs. Next, in Table 9(c), we study the performance when adding different numbers of LoRA layers to the pre-trained model, where we observe that 12 blocks achieve slightly better results. In the next study (Table 9(d)), we investigate the performance of our method by adding learnable LoRA layers to both encoders, effectively incorporating synchronous training. This variant of our method shows a drop of 1.37% in the final accuracy, showing the importance of our proposed asynchronous teacher. We then study different parameters for the controlled

Table 8: Ablation study on all datasets. Here, Asy. Cont., Cal. Tun., Con. tun. refers to asynchronous contrastive tuning, controlled fine-tuning and consistency-guided incremental tuning. The reported values are average and standard deviations across three runs.

| Asy. Cont. | Cal. Tun. | Con. Tun. | Caltech101 | CIFAR100 | Country211 | CUB200 | DTD | EuroSat | FGVCAircraft | Food101 |
|---|---|---|---|---|---|---|---|---|---|---|
| ✓ | ✓ | ✓ | $86.86_{\pm0.03}$ | $78.31_{\pm0.02}$ | $7.42_{\pm0.04}$ | $77.38_{\pm0.01}$ | $55.11_{\pm0.05}$ | $62.25_{\pm0.03}$ | $18.98_{\pm0.02}$ | $71.59_{\pm0.08}$ |
| ✓ | ✗ | ✓ | $85.89_{\pm0.05}$ | $78.45_{\pm0.03}$ | $7.45_{\pm0.02}$ | $77.27_{\pm0.04}$ | $54.59_{\pm0.01}$ | $62.91_{\pm0.05}$ | $18.20_{\pm0.03}$ | $71.83_{\pm0.07}$ |
| ✓ | ✓ | ✗ | $86.01_{\pm0.02}$ | $78.96_{\pm0.01}$ | $7.46_{\pm0.03}$ | $77.20_{\pm0.05}$ | $54.82_{\pm0.02}$ | $62.31_{\pm0.01}$ | $18.91_{\pm0.09}$ | $71.47_{\pm0.03}$ |
| ✓ | ✗ | ✗ | $84.62_{\pm0.12}$ | $78.37_{\pm0.04}$ | $7.49_{\pm0.01}$ | $76.44_{\pm0.03}$ | $54.83_{\pm0.05}$ | $62.22_{\pm0.02}$ | $18.23_{\pm0.01}$ | $71.39_{\pm0.04}$ |
| ✗ | ✗ | ✗ | $83.47_{\pm0.10}$ | $75.58_{\pm0.02}$ | $7.42_{\pm0.05}$ | $74.10_{\pm0.01}$ | $54.72_{\pm0.03}$ | $62.20_{\pm0.04}$ | $15.51_{\pm0.02}$ | $71.09_{\pm0.05}$ |

| Asy. Cont. | Cal. Tun. | Con. Tun. | MiniImageNet | Flowers102 | OxfordPets | Resisc-45 | StanfordCars | SUN397 | VOC 2007 | GTSRB | Average |
|---|---|---|---|---|---|---|---|---|---|---|---|
| ✓ | ✓ | ✓ | $95.34_{\pm0.03}$ | $98.18_{\pm0.02}$ | $87.79_{\pm0.01}$ | $62.62_{\pm0.04}$ | $24.40_{\pm0.05}$ | $66.11_{\pm0.03}$ | $64.45_{\pm0.02}$ | $26.05_{\pm0.03}$ | 61.43 |
| ✓ | ✗ | ✓ | $95.04_{\pm0.01}$ | $98.01_{\pm0.03}$ | $88.25_{\pm0.02}$ | $61.77_{\pm0.05}$ | $24.14_{\pm0.01}$ | $66.01_{\pm0.04}$ | $64.89_{\pm0.03}$ | $26.29_{\pm0.04}$ | 61.31 |
| ✓ | ✓ | ✗ | $95.34_{\pm0.02}$ | $98.15_{\pm0.01}$ | $87.92_{\pm0.03}$ | $61.31_{\pm0.02}$ | $24.07_{\pm0.04}$ | $65.98_{\pm0.05}$ | $64.59_{\pm0.01}$ | $26.05_{\pm0.02}$ | 61.28 |
| ✓ | ✗ | ✗ | $95.34_{\pm0.04}$ | $98.09_{\pm0.05}$ | $88.10_{\pm0.01}$ | $57.21_{\pm0.03}$ | $23.18_{\pm0.02}$ | $65.94_{\pm0.01}$ | $64.72_{\pm0.04}$ | $18.02_{\pm0.09}$ | 60.26 |
| ✗ | ✗ | ✗ | $95.31_{\pm0.05}$ | $98.09_{\pm0.02}$ | $88.10_{\pm0.04}$ | $49.83_{\pm0.01}$ | $18.90_{\pm0.03}$ | $65.68_{\pm0.02}$ | $64.72_{\pm0.05}$ | $15.53_{\pm0.08}$ | 58.77 |

tuning step, specifically the learning rate factor $C_f$, fine-tuning layers $C_l$, and fine-tuning start epoch $E_c$, in Table 9(e-g). The results from this study show the best performance when we fine-tune the pre-trained layers with a higher LR factor, effectively fine-tuning the pre-trained layers with lower LR and allowing smaller changes in the pre-trained weights. For $C_l$, we find the best results when we fine-tune half of the layers (6 out of 12) of the pre-trained encoder; training just the final layer or the whole network results in reduced accuracy. Moreover, the best result is obtained when starting the fine-tuning step after 10 epochs of training of the newly added LoRA layers. Table 9(h) shows the results for training different layers during incremental learning, where tuning both the encoder (last few layers) and the LoRA shows the best results. Table 9(i) shows the study on using the encoder from the first session vs. the encoder from the most recent session to enforce consistency. As we find from this table, the student encoder from the first session results in a higher performance. We also study the performance of two new variants of LoRA, including qLoRA (Dettmers et al., 2024) and LoRA+ (Hayou et al., 2024) in Table 9(j). As we find from this table, LoRA+ shows a minor improvement over the Naive LoRA, while qLoRA results in a slight drop in performance. However, in our work, we used the default LoRA method given its popularity and wide use in the area as the default approach to fine-tuning.

## 4.5 Computational complexity

Finally, we analyze the computational complexity of our method compared to the previous SOTA, PriViLege (Park et al., 2024). Our analysis is presented in Table 9(k). In terms of the number of parameters, CoACT has significantly fewer parameters than PriViLege (86 million vs. 149 million) since CoACT is based on a vision-only backbone, whereas PriViLege utilizes a vision-language backbone. Additionally, CoACT achieves higher throughput than PriViLege during both training and inference. Specifically, PriViLege achieves a training throughput of 617 samples/second, whereas CoACT reaches 930. Similarly, the inference throughput is 1,236 for PriViLege and 1,850 for CoACT.

Table 9: Ablation of different parameters of CoACT on FSCIL averaged across 16 datasets, including different alternate for LoRA, training epochs, number of LoRA layers, settings for adding LoRA to the encoders, different parameters for controlled tuning (learning rate factor, fine-tuning layers, and fine-tuning epochs), and different tuning layers.



(a) ACT

| Module | Accuracy |
|--------|----------|
| LoRA | 60.15 |
| Adapter | 60.01 |
| Prompt | 59.87 |

(b) Epochs

| Epochs | Accuracy |
|--------|----------|
| 10 | 58.98 |
| 20 | 59.81 |
| 50 | 60.15 |

(c) LoRA layers

| Layers | Accuracy |
|--------|----------|
| 3 | 59.92 |
| 6 | 60.01 |
| 12 | 60.26 |

(d) Async. encoder

| Ablation | Accuracy |
|----------|----------|
| Asyn. enc. | 60.26 |
| Same enc. | 58.89 |



(e) LR factor ($C_f$)

| $C_f$ | Accuracy |
|-------|----------|
| 1.0 | 59.4 |
| 0.5 | 60.37 |
| 0.1 | 61.14 |

(f) Ft. layers ($C_l$)

| $C_l$ | Accuracy |
|-------|----------|
| All | 60.51 |
| Half | 61.14 |
| Last | 60.76 |

(g) Ctr. epochs ($E_c$)

| $E_c$ | Accuracy |
|-------|----------|
| 0 | 60.64 |
| 10 | 61.21 |
| 25 | 60.62 |

(h) Tuning layers

| Tuning layer | Accuracy |
|--------------|----------|
| Encoder only | 61.22 |
| LoRA only | 60.94 |
| Enc. + LoRA | 61.28 |

(i) Incr. consistency

| Setup | Accuracy |
|-------|----------|
| Consistency w/ first | 61.43 |
| Consistency w/ last | 61.22 |

(j) Different LoRA

| Setup | Accuracy |
|-------|----------|
| LoRA | 58.96 |
| qLoRA | 58.92 |
| LoRA+ | 59.11 |

(k) Compute complexity

| Method | Param. | Thr. (samp./sec.) Train | Infer. |
|--------|--------|-------|--------|
| PriViLege | 149M | 617 | 1236 |
| CoACT | 86M | 930 | 1850 |

## 5 Conclusion

To enable few-shot class-incremental learning with pre-trained large vision models, we propose CoACT. Our method can effectively tune a foundation model to learn new classes without losing the generalization of the pre-training or forgetting previously learned classes. Extensive studies show the effectiveness of our method, achieving higher accuracy, lower forgetting, and robustness in low-shot settings. Notably, CoACT outperforms prior works by up to 5.02% in standard FSCIL setup and by up to 12.51% on FSCIT in the individual datasets. We present comprehensive experiments on different components of CoACT.

**Limitations.** Our study only focuses on class-incremental learning, but the notion of tuning a foundation model can be explored with other forms of continual learning, such as task-incremental learning. Additionally, a limitation of CoACT is that it comprises a few hyper-parameters, although our study shows a small sensitivity to those parameters in our study across 16 datasets.

**Broader impact.** Our work focuses on the few-shot tuning of foundation models with impressive generalization capability to effectively learn new classes with limited data. While our method does not have any direct negative impact, it comes with the same potential risk of being misused as any tuning method, where a model could be tuned to learn unwanted use cases. On the other hand, the proposed solution can potentially have a large positive impact, as the idea proposed in this work can be explored to update foundation models for new classes without the need to train from scratch, saving compute and other resources.

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
