# OpenReview forum: "Consistency-Guided Asynchronous Contrastive Tuning for Few-Shot Class-Incremental Tuning of Foundation Models"
_TMLR — Accepted by TMLR_

### Review · Reviewer_3t7d · 2025-02-05

**Summary Of Contributions:**

This paper mainly targets few-shot class-incremental learning/tuning especially for foundation models. The authors propose CoACT, which includes three components, i.e. asynchronous contrastive tuning, controlled fine-tuning, and consistency-guided incremental tuning, to avoid the problem of forgetting learned sessions. The authors have conducted extensive experiments on multiple datasets to validate the effectiveness of the proposed framework.

**Audience:**

Yes

**Claims And Evidence:**

Yes

**Requested Changes:**

Please refer to the weaknesses.

**Strengths And Weaknesses:**

Strength:
1. The authors study a very practical setting about leveraging foundation models for few-shot class-incremental learning.
2. The design of proposed modules is reasonable.
3. The paper is clearly written.

Weaknesses:
1. I wonder how prompt is implemented for Tab.9(a). Is it the same as visual prompt tuning?
2. More foundation model backbones can be adopted in experiments.
3. It would be better to show the computational cost.
4. The authors can discuss whether such a framework can be extended to other tasks such as detection and segmentation.

---

> ### Author Response · Authors · 2025-02-23
>
> We thank the reviewer for their valuable feedback and comments. Below, we address each point and highlight the related changes in the revised manuscript with the color **blue**.
>
> >  I wonder how prompt is implemented for Tab.9(a). Is it the same as visual prompt tuning?
>
> In Table 9(a), the prompt implementation follows the visual prompt tuning (VPT) (Jia et al., 2022) approach, where learnable prompts are added to the input embeddings of the vision transformer. We have now added this description to Section 4.4, paragraph 2.
>
>
>
> > More foundation model backbones can be adopted in experiments.
>
> As per the comment, we have now explored more foundation model backbones, including SwinTransformers (Swin-B), DINO ViT-B/14, and the vision encoder from CLIP ViT-B/16, and included them in Table 7 where we previously provided results for ViT-B/16, ViT-B/32, and ViT-L/14. Similar to our findings with the larger encoders, CoACT generalizes across backbones and shows strong performances on Swin-B, DINO ViT-B/14, and CLIP ViT-B/16. This is added to the second to last paragraph of page 10.
>
>
> > It would be better to show the computational cost.
>
> In the revised manuscript, we have now included a discussion on the computation complexity of our proposed method, and its comparison to the previous SOTA, PreViLege. Specifically, we have added the following discussion to page 13 (new Section 4.5):
>
> Finally, we analyze the computational complexity of our method compared to the previous SOTA, PriViLege (Park et al., 2024). Our analysis is presented in Table 9(k). In terms of the number of parameters, CoACT has significantly fewer parameters than PriViLege (86 million vs. 149 million) since CoACT is based on a vision-only backbone, whereas PriViLege utilizes a vision-language backbone. Additionally, CoACT achieves higher throughput than PriViLege during both training and inference. Specifically, PriViLege achieves a training throughput of 617 samples/second, whereas CoACT reaches 930. Similarly, the inference throughput is 1,236 for PriViLege and 1,850 for CoACT.
>
> | Method     | Parameters | Train Throughput | Inference Throughput |
> |------------|------------|------------------|----------------------|
> | PriViLege  | 149M       | 617              | 1,236                |
> | CoACT      | 86M        | 930              | 1,850                |
>
>
> > The authors can discuss whether such a framework can be extended to other tasks such as detection and segmentation.
>
> While the extension of CoACT to dense prediction tasks such as detection and segmentation is an interesting direction, we believe it is beyond the scope of the current work, primarily due to the specialized nature of these tasks. Dense prediction tasks, unlike classification, require models to produce pixel- or region-level outputs, which introduces additional complexities in both architecture design and training pipelines. For instance, detection tasks often rely on region proposal networks or anchor-based methods, while segmentation tasks typically employ encoder-decoder structures with skip connections to capture fine-grained spatial details. These tasks also demand task-specific loss functions, such as bounding box regression for detection or pixel-wise cross-entropy for segmentation, which are fundamentally different from the classification losses used in our current framework. Adapting CoACT to these tasks would require significant modifications to the framework, such as rethinking how consistency-guided tuning operates on dense outputs or designing task-specific adaptations for LoRA modules.

---

### Review · Reviewer_BZ3w · 2025-02-06

**Summary Of Contributions:**

This paper focuses on the Few-Shot Class-Incremental Learning (FSCIL) task and proposes a Consistency-guided Asynchronous Contrastive Tuning (CoACT) method by using three experiential tuning components. In addition, a new paradigm of continual learning called Few-shot Class-Incremental Tuning (FSCIT) is also proposed. Extensive experiments have been conducted to support the proposed CoACT method.

**Audience:**

Yes

**Broader Impact Concerns:**

No, there is no concern on the ethical implications of the work.

**Claims And Evidence:**

Yes

**Requested Changes:**

Please kindly refer to the above weaknesses.

**Strengths And Weaknesses:**

**Strengths**:
1. The paper is well organized, which is easy to follow.
2. The new setting of Few-shot Class-Incremental Tuning (FSCIT) proposed is interesting and practical in the real applications.
3. The experimental results are good compared with the baselines.


**Weaknesses**:
1. The main concern is about the novelty of the proposed CoACT method. In fact, all the components of CoACT are common and widely used in the existing works in the literature. The motivation of each component is very experiential. Therefore, the novelty and contribution of this work are recommended to be further justified.
2. What is the main difference between $L_{ACL}$ and $L_{CR}$? It seems that they are totally the same.
3. Another concern is that in the experiment section, the proposed method uses a different pre-trained backbone from other comparison methods, such as SV-T, CPE-CLIP and PriViLege. This is not very fair, making the experimental results not convincing. In addition, the pre-trained backbone in the proposed method is pre-trained on ImageNet-21K. What are the pre-trained datasets used in the comparison methods?
4. Note that because the pre-trained dataset is ImageNet-21K, how to avoid the data leakage problem? We can see that the proposed method has achieved a really high average accuracy of 96.24% on the miniImageNet dataset, which may be caused by the data leakage problem.
5. In the ablation study in Table 8, there are no significant differences between the first three rows. How to demonstrate the effectiveness of the controlled fine-tuning and consistency-guided incremental tuning components?
6. In fact, many LoRA variants have been proposed in recent years. How does the proposed method compare to the latest LoRA variants?

---

> ### Author Response · Authors · 2025-02-23
>
> We thank the reviewer for their valuable feedback and comments. Below, we address each point and highlight the related changes in the revised manuscript with the color **blue**.
>
> > The main concern is about the novelty of the proposed CoACT method. In fact, all the components of CoACT are common and widely used in the existing works in the literature. The motivation of each component is very experiential. Therefore, the novelty and contribution of this work are recommended to be further justified.
>
> While some components of CoACT are inspired by existing works, the novelty of our method lies in the unified framework specifically designed for few-shot class-incremental learning (FSCIL). Each component is carefully integrated to address the unique challenges of FSCIL, and they also contain novelty of their own:
>
> **Asynchronous Contrastive Tuning (ACL):**
> ACL enforces consistency between a student encoder (with learnable LoRA modules) and a teacher encoder (updated via EMA) to balance adaptability to new classes and retention of the pre-trained model’s generalization. The key novelty lies in the asynchronous design, where the teacher is initialized from the pre-trained model and updated slowly via EMA, ensuring stable predictions while the student adapts to new classes. This approach prevents overfitting in few-shot settings, which is a critical challenge in FSCIL. **No prior works have proposed contrastive learning in such a setting.**
>
> **Controlled Fine-Tuning:**
> This component introduces a two-step training protocol for the first session: first training the LoRA modules with a high learning rate, followed by fine-tuning the last few layers of the pre-trained model with a lower learning rate. Here, this strategy ensures that the model adapts to new classes without drastically altering the pre-trained weights, preserving the generalization capabilities of the foundation model. While this technique builds upon well-known concepts like partial fine-tuning, **its role in continual fine-tuning in a few-shot learning setup has never been explored.**
>
> **Consistency-Guided Incremental Tuning:**
> This component is a variant of contrastive learning tailored for incremental sessions. It enforces consistency between the predictions of the learnable encoder and a frozen encoder from the first session. The specific goal is to prevent forgetting of previously learned classes while maintaining the generalization of the pre-trained model. **This approach is novel in its use of a frozen encoder from the first session, which acts as a stable reference point for regularization across all incremental sessions.**
>
> Overall, CoACT introduces a novel method that significantly reduces forgetting (as shown in Figure 3, left) and improves performance over existing methods in the FSCIL setup. Additionally, another key contribution of our work is the exploration of a new, more challenging, and practical direction, i.e., Few-Shot Class-Incremental Tuning (FSCIT), where we do not assume the availability of a large supervised base session, making it more applicable to real-world scenarios.
>
>
>
> > What is the main difference between $L_{ACL}$ and $L_{CR}$ It seems that they are totally the same.
>
> While $L_{ACL}$ and $L_{CR}$ are similar in formulation, the main differences between these two functions are the input and the role they play. Specifically, $L_{ACL}$ enforces consistency between the student encoder (with LoRA) and the EMA teacher encoder during the first session to balance adaptability and generalization, while $L_{CR}$ ensures consistency between the current student encoder and a frozen encoder from the first session in subsequent sessions to prevent forgetting.

---

> > ### Author Response · Authors · 2025-02-23
> >
> > > Another concern is that in the experiment section, the proposed method uses a different pre-trained backbone from other comparison methods, such as SV-T, CPE-CLIP and PriViLege. This is not very fair, making the experimental results not convincing. In addition, the pre-trained backbone in the proposed method is pre-trained on ImageNet-21K. What are the pre-trained datasets used in the comparison methods?
> >
> >
> > For our experiments in Tables 2, 3, and 4, we closely followed the convention of prior works in the area of FSCIL (Tao et al., 2020, Chi et al., 2022, FACT (Zhou et al., 2022a, Song et al., 2023, Park et al., 2024, Qiu et al., 2023, and others). In all the prior works mentioned in these tables, different backbones have been used. For example, CPE-CLIP and PriViLege use the CLIP encoder, SV-T uses SwinTransformer, and earlier methods (e.g., FACT, LIMIT) use ResNet encoders.
> >
> > In our case, CoACT focuses on tuning a vision-only pre-trained encoder and performs the main experiments on the most common and basic variant of the transformer, ViT-B. In comparison, SV-T utilizes a specialized transformer called SwinTransformer, and more recent methods like CPE-CLIP, and PriViLege use much larger vision-language models (e.g. CLIP).
> >
> > In terms of the training data, ViT-B used in our proposed solution (and SwinTransformer in SV-T) is pre-trained on 14.2M samples. In contrast, the previous SOTA PriViLege uses a CLIP encoder pre-trained on 400M samples. Additionally, ViT-16 consists of only 86M parameters compared to 149M parameters of the previous SOTA PreViLege (please refer to Table 9 (k) of the revised manuscript).
> >
> > Having said the above, we design CoACT as a model-agnostic solution which can be utilized to fine-tune various existing foundation models. In Table 7, we have reported the performance of CoACT with other encoders, and to further address your comment, we have now added three more backbones to this study, Swin-B, DINO ViT-B/14, and CLIP ViT-B/16.
> >
> >
> >
> > > Note that because the pre-trained dataset is ImageNet-21K, how to avoid the data leakage problem? We can see that the proposed method has achieved a really high average accuracy of 96.24\% on the miniImageNet dataset, which may be caused by the data leakage problem.
> >
> >
> > Thank you for this important question. Kindly note that we have used 16 datasets of diverse concepts to evaluate our method, one of which is the miniImageNet dataset in question. These datasets consist of diverse concepts including fine-grained object classification, texture recognition, scene recognition, traffic sign recognition, and satellite image, most of which are very different from the pre-training dataset ImageNet-21K, and our proposed method consistently demonstrates strong performances across the dataset. We can observe from Table 6 that while the miniImageNet results are indeed strong, there are other datasets which show even higher performances, for instance, Flowers102. We believe the strong performance on miniImageNet is a result of, not data leakage, but rather the closeness of the training and test 'domains'.
> > In the case of miniImagenet, we followed the convention of using an encoder pre-trained on the 'train' set of ImageNet-21K and we tested on the 'validation' set of miniImageNet. Prior works such as (Qui et al. 2023) have used this protocol, and we have no reason to believe any data leakage exists as the test set of miniImageNet is a subset of the test set of ImageNet-21K, and the test set of ImageNet-21K has not been used during training.
> > Regardless, we have now added additional experiments on different pre-trained backbones (Table 7 of the revised manuscript), including, the vision encoder of CLIP ViT-B/16 trained on 400M pairs of images and captions, and DIVO ViT-B/14 pre-trained on 142M samples in a self-supervised manner. Our method shows strong performance for all the encoders (even when not trained on the ImageNet-21K dataset) suggesting strong generalizability of our solution.
> >
> >
> >
> >
> >
> > > In the ablation study in Table 8, there are no significant differences between the first three rows. How to demonstrate the effectiveness of the controlled fine-tuning and consistency-guided incremental tuning components?
> >
> > While removing one component at a time (e.g., controlled fine-tuning or consistency-guided incremental tuning) shows a relatively small drop in performance, removing **both components simultaneously** results in a significant performance **drop of 1.17\%** (from 61.43\% to 60.26\%).
> > This improvement, in comparison to the enhancements obtained by prior work, is indeed considerable, which highlights the importance of these components in the proposed method.
> > Additionally, when all three components are removed, the performance drops further to 58.77\%, demonstrating the cumulative impact of each component. A detailed discussion is provided in Section 4.4.

---

> > > ### Author Response · Authors · 2025-02-23
> > >
> > > > In fact, many LoRA variants have been proposed in recent years. How does the proposed method compare to the latest LoRA variants?
> > >
> > > As per the comment, we have now explored two new variants of LoRA, including qLoRA (Dettmers et al., 2024) and LoRA+ (Hayou et al. 2024). The results are presented in the table below (new Table Table 9 (j) in the paper). As we find from these experiments, LoRA+ shows some improvement over the Naive LoRA, while qLoRA results in a slight drop in performance. Overall, CoACT considerably outperforms all methods by a considerable margin. However, in our work, we used the default LoRA method given its popularity and wide use in the area as the default approach to fine-tuning. We have now added this discussion at the end of Section 4.4.
> > >
> > > | Method | Avg. Acc |
> > > |-------|-----------|
> > > | CoACT | 61.43 |
> > > | LoRA  | 58.96 |
> > > | qLoRA | 58.92 |
> > > | LoRA+ | 59.11 |

---

### Review · Reviewer_dsg8 · 2025-02-10

**Summary Of Contributions:**

The paper introduces a new method, **Consistency-Guided Asynchronous Contrastive Tuning (CoACT)**, for incrementally tuning pretrained vision foundation models (specifically ViT models) to learn new classes in few-shot settings. The method consists of three main components:

1. **Asynchronous Contrastive Tuning**: This involves tuning the ViT model using LoRA adapters for new classes while applying a supervised contrastive loss with an exponential moving average (EMA) teacher. This ensures consistency with the pretrained model.
2. **Controlled Fine-Tuning**: The method first fine-tunes the ViT model using LoRA adapters and then, after a few epochs, switches to full fine-tuning of the linear layers in the last $C_l$ transformer blocks.
3. **Consistency-Guided Incremental Tuning**: In addition to asynchronous contrastive tuning, this component uses an extra supervised contrastive loss to enforce consistency with the frozen encoder after the first learning session (i.e., after learning the initial set of classes).

The authors evaluate their approach on two settings: **Few-Shot Class-Incremental Learning (FSCIL)** and **Few-Shot Class-Incremental Tuning (FSCIT)**. Unlike FSCIL, FSCIT does not involve a base class training stage, which is common in few-shot learning literature. The method is tested on multiple datasets, demonstrating strong performance.

**Audience:**

Yes

**Claims And Evidence:**

No

**Requested Changes:**

**Conclusion:**

The paper presents an effective method for few-shot class-incremental learning, supported by strong experimental results. However, the motivation behind several design choices is unclear, and as a result, the paper does not provide substantial insights into the problem. Additionally, further experiments are needed to fully validate the method’s generalizability, particularly with different pretrained ViT models. Addressing these concerns would significantly strengthen the paper and its contributions to the field.

**Strengths And Weaknesses:**

**Strengths:**

1. The paper presents extensive experiments and thorough ablation studies, which provide a detailed evaluation of the method. However, as noted in the weaknesses section, some experiments are missing.
2. The proposed method achieves strong results, outperforming several strong baselines and prior approaches.
3. The paper is generally well-written, though the motivation and intuition behind the design choices is unclear (discussed further in the weaknesses section).


**Weaknesses:**

1. **Unclear Intuition for Design Choices**:
   - The use of an EMA teacher in the asynchronous contrastive tuning component is not well-justified. If the student parameters $\theta$ in the EMA formula ($\theta^{\prime\prime} = m \cdot \theta^{\prime\prime} + (1 - m) \cdot \theta$) are the frozen pretrained parameters (without LoRA adapters), the teacher parameters $\theta^{\prime\prime}$ would always equal the frozen pretrained parameters $\theta$. This raises questions about the role and necessity of the EMA teacher. Why are the LoRA adapters excluded from the EMA updates?
   - The controlled fine-tuning strategy is also unclear. Starting with LoRA adapters and then switching to full fine-tuning of the linear layers in the last $C_l$ transformer blocks seems counter-intuitive and lacks a clear motivation. Additionally, some clarifications are needed: Are both LoRA adapters and full fine-tuning applied to the same linear layers? Is this strategy used only in the first session or in all sessions?
   - In the consistency-guided incremental tuning component, it is unclear why consistency is enforced with the frozen encoder from the first learning session rather than the previous session (e.g., using the frozen encoder from session $t-1$ for session $t$). The latter approach might better prevent forgetting across all previous sessions. Furthermore, it is unclear which frozen encoder is used—the teacher or the student?

2. **Lack of Deep Insight**:
   - The paper appears to combine several tricks to boost performance, but the role and significance of each component are not deeply analyzed or understood. As a result, the paper does not provide substantial insights into the problem of few-shot class-incremental learning.

3. **Missing Experiments: The role of the pretrained ViT model is not fully explored.**:
   -  Prior methods like PriViLege and CPE-CLIP use CLIP’s ViT-B/16 model, while this method uses a ViT-B/16 model pretrained on ImageNet-21k. The authors should demonstrate that their method works well with CLIP’s ViT-B/16 model as well, to show that the improvements are not simply due to choice of the pretrained model. Since their approach is backbone-agnostic, this should be straightforward.
   - More broadly, it would be interesting to see how the method performs with other pretrained ViT models, such as DINOv2’s ViT models (a self-supervised foundation model) and CLIP’s ViT models. It is important to demonstrate whether the improvements observed with the ImageNet-21k pretrained ViT model generalize to other pretrained models.

4. **Significance of Performance Differences in Ablation Study**:
   - In Table 8 (ablation study of the three components), the differences between the top three rows are very small. It would be helpful to report the standard deviation across multiple runs (the authors mention using three runs) to determine if these differences are statistically significant.

---

> ### Author Response · Authors · 2025-02-23
>
> We thank the reviewer for their valuable feedback and comments. Below, we address each point and highlight the related changes in the revised manuscript with the color **blue**.
>
> > The use of an EMA teacher in the asynchronous contrastive tuning component is not well-justified. Why are the LoRA adapters excluded from the EMA updates?
>
> The use of an Exponential Moving Average (EMA) teacher in the asynchronous contrastive tuning component of CoACT is designed to provide a stable reference for the student encoder. The primary motivation for excluding LoRA adapters from the EMA updates is to prevent rapid fluctuations in the teacher encoder. The teacher's role is to serve as a slow-moving reference point that mitigates overfitting and retains the generalization capabilities of the pre-trained foundation model. If LoRA adapters were included in the EMA updates, the teacher encoder would change more dynamically, reducing its ability to regularize the student effectively. This would undermine the core objective of asynchronous contrastive learning, which is to balance adaptability with generalization retention.
>
> By design, the teacher encoder is initialized as the pre-trained model and updated using the EMA of the student encoder. The update rule is given by:
> $\theta'' = m \cdot \theta'' + (1 - m) \cdot \theta$
> where $\theta$ represents the student encoder’s learnable (last $C_l$ layer; Section 3.2.2) parameters (excluding LoRA), $\theta''$ represents the teacher encoder’s parameters, and $m$ is the momentum parameter. Since LoRA adapters are only applied to the student encoder, their parameters are not included in this EMA formulation. This ensures that the teacher encoder remains largely aligned with the pre-trained model while still gradually incorporating useful updates from the student encoder in a controlled manner.
>
> Another reason for excluding LoRA adapters from the EMA teacher is to mitigate overfitting in few-shot learning settings. Since few-shot learning involves a very limited number of labelled examples, models tend to overfit quickly when fine-tuned. By maintaining a teacher that remains close to the pre-trained model, CoACT ensures that the student does not drift too far from the foundation model’s strong generalization capabilities. The teacher encoder acts as a form of regularization, constraining the student's learned representations and preserving knowledge from the pre-trained model.
>
> As later shown in Table 9(d), without the asynchronous design the performance drops by 1.37\%. We have further clarified a summary of the reason for excluding LoRA from the teacher update in Section 3.2.1.
>
>
> > The controlled fine-tuning strategy is also unclear. Starting with LoRA adapters and then switching to full fine-tuning of the linear layers in the last transformer blocks seems counter-intuitive and lacks a clear motivation. Additionally, some clarifications are needed: Are both LoRA adapters and full fine-tuning applied to the same linear layers? Is this strategy used only in the first session or in all sessions?
>
>
> Since the newly added LoRA modules are randomly initialized, we begin by training only the LoRA modules of the student encoder with a higher learning rate for the initial $E_c$ epochs of training to prevent the propagation of randomness to the well-trained pre-trained encoder, which could otherwise lead to forgetting of its out-of-the-box general knowledge. This is followed by a fine-tuning stage where the last $C_l$ layers of the pre-trained encoder are fine-tuned with a reduced LR (scaled by a factor of $C_f$). We focus on fine-tuning only the last $C_l$ layers, as the later layers of a pre-trained model are responsible for learning domain-specific fine-grained features, whereas the earlier layers are more general and transferable to a wide range of tasks (Neyshabur et al., 2020). Importantly, this fine-tuning is performed under the consistency constraint of asynchronous contrastive learning, ensuring the model learns target-domain features effectively without overfitting or forgetting pre-trained knowledge. **This balance between adaptability and generalization enables effective adaptation to new classes while preserving the pre-trained encoder’s foundational capabilities.**
>
> As for the question about whether we apply LoRa and full fine-tuning to the same layer, the answer is yes - some (last $C_l$) of the layers of the pre-trained encoders are tuned with both LoRA and fine-tuning. Please kindly refer to Section 4.4, Table 9 (f), where we present the results of our solution when the last half of the encoder (layers 7-12) are fine-tuned, along with LoRA being applied to all layers. The two-setup controlled fine-tuning strategy is applied during the first session, while in later sessions, fine-tuning continues with LoRA across all layers, along with fine-tuning the last $C_l$ layers.
>
> We have further added a summary of these discussions to Section 3.2.2.

---

> > ### Author Response · Authors · 2025-02-23
> >
> > > In the consistency-guided incremental tuning component, it is unclear why consistency is enforced with the frozen encoder from the first learning session rather than the previous session (e.g., using the frozen encoder from session $t-1$ for session $t$). The latter approach might better prevent forgetting across all previous sessions. Furthermore, it is unclear which frozen encoder is used—the teacher or the student?
> >
> >
> > Thank you for this insightful question. To answer the second question first, we enforce consistency with the frozen **student** encoder since the student learns the target distribution while the teacher only acts as a stable reference to guide the learning.
> > Second, we use the student encoder from the **first** learning session rather than the most recent session (e.g., instead of using the frozen encoder from session $t-1$ for session $t$) since our approach is better equipped to prevent the model from gradually drifting away (with cumulative overfitting over the sessions) from the pre-trained encoder’s knowledge, which prevents increased forgetting of the foundational generalization capabilities. By anchoring the consistency to the first session’s encoder, CoACT effectively balances the learning of new classes with the retention of pre-trained knowledge, mitigating catastrophic forgetting. We have further clarified this aspect of our method in Section 3.2.3 by including a summary of this discussion. However, as per your suggestion, we have now included an experiment (Table 9 (i)) to compare the performance of our approach with the approach of using the most recent student encoder, where our approach shows relatively higher accuracy.
> > In a standard class-incremental learning setup, the encoder updates across multiple sessions as
> >     $f_{\theta_t} = f_{\theta^{t-1}} + \Delta_{\theta_t}$
> > where $\Delta_{\theta_t}$ represents the parameter updates due to learning new classes at session $t$. If is consistency enforced with the encoder from the previous session, $t-1$. As a result, the model incrementally adapts but can also gradually drift away from its initial state, leading to
> >     $f_{\theta_T} = f_{\theta^0} + \sum_{t=1}^{T} \Delta_{\theta_t}$.
> > This, in turn, can result in a gradual loss of generalization, where the model drifts away from the foundational encoder.
> > Instead, enforcing consistency with the frozen encoder from the first session $f_{\theta^0}$ prevents progressive divergence. This ensures that at each session,
> >     $f_{\theta_t}=f_{\theta^0}+\Delta_{\theta_t}'$, where $\| \Delta_{\theta_t}' \| \ll \| \sum_{i=1}^{t} \Delta_{\theta_i} \|$.
> >
> >
> > > Lack of Deep Insight:
> > The paper appears to combine several tricks to boost performance, but the role and significance of each component are not deeply analyzed or understood. As a result, the paper does not provide substantial insights into the problem of few-shot class-incremental learning.
> >
> > We would like to clarify that CoACT is not merely a combination of performance-boosting techniques. Rather, it introduces a novel method consisting of three carefully designed components, each aimed at addressing the challenges of continuous learning in a few-shot setting (for a detailed discussion on the intuition behind each component, please kindly refer to paragraph 2 of the Introduction).  To provide a comprehensive understanding of the role and significance of each component we provide an ablation study in Table 8, which highlights their individual and collective impact. Additionally, Table 9 presents detailed studies on different components of CoACT, analyzing various design choices and their effects on model performance.
> >
> >
> >
> > > Missing Experiments: The role of the pre-trained ViT model is not fully explored.
> > Prior methods like PriViLege and CPE-CLIP use CLIP’s ViT-B/16 model, while this method uses a ViT-B/16 model pretrained on ImageNet-21k. The authors should demonstrate that their method works well with CLIP’s ViT-B/16 model as well, to show that the improvements are not simply due to choice of the pretrained model. Since their approach is backbone-agnostic, this should be straightforward.
> > More broadly, it would be interesting to see how the method performs with other pretrained ViT models, such as DINOv2’s ViT models (a self-supervised foundation model) and CLIP’s ViT models. It is important to demonstrate whether the improvements observed with the ImageNet-21k pretrained ViT model generalize to other pretrained models.
> >
> >
> >
> > As per your suggestion, we have now explored additional encoders and reported the results by expanding Table 7 in the revised manuscript. Specifically, we present the results for DINOv2's ViT model, CLIP's ViT model, and SwinTransformer model. Similar to findings on larger encoders (Table 7), CoACT generalizes across backbones and shows strong performance on Swin-B, DINO ViT-B/14, and CLIP ViT-B/16. In the revised manuscript, we have revised the discussion in the second to last paragraph of Section 4.3.

---

> > > ### Author Response · Authors · 2025-02-23
> > >
> > > > Significance of Performance Differences in Ablation Study
> > > In Table 8 (ablation study of the three components), the differences between the top three rows are very small. It would be helpful to report the standard deviation across multiple runs (the authors mention using three runs) to determine if these differences are statistically significant.
> > > Requested Changes:
> > >
> > >
> > > As per your suggestion, we have now revised Table 8 to show the standard deviations over three runs for each dataset and ablation version of the model. As we find from the revised table, the standard deviations are quite small with respect to the reported average values over three runs. Additionally, please note that while removing one component at a time (e.g., controlled fine-tuning or consistency-guided incremental tuning) shows a relatively small drop in performance, removing **both components simultaneously** results in a significant performance **drop of 1.17\%** (from 61.43\% to 60.26\%). This improvement, in comparison to the enhancements obtained by prior work, is indeed considerable, which highlights the importance of these components in the proposed method. Additionally, when all three components are removed, the performance drops further to 58.77\%, demonstrating the cumulative impact of each component.

---

### Author Response · Authors · 2025-02-23
**Rebuttal Summary**

We sincerely thank the reviewers for their time and for providing detailed and constructive feedback. We are happy to see the overall engaging comments given by all the reviewers. We are glad to see that reviewers find the paper well-organized and clearly written (Reviewers BZ3w, 3t7d), appreciate the novel and practical setting of Few-Shot Class-Incremental Tuning (FSCIT) (Reviewer BZ3w), and acknowledge the strong experimental results and extensive ablation studies that demonstrate the effectiveness of the proposed method (Reviewers dsg8, BZ3w). We have carefully addressed all the concerns raised by the reviewers under the individual response section. Following, we provide a summary of our responses, and the changes we have made to the manuscript.

— Clarification of design choices: We have provided detailed justifications for the use of the EMA teacher, controlled fine-tuning strategy, and consistency-guided incremental tuning, addressing concerns about the motivation and role of each component (Reviewer dsg8).

— Additional backbone experiments: We expanded the evaluation to include additional foundation model backbones, such as SwinTransformers, DINO ViT-B/14, and CLIP ViT-B/16, to demonstrate the generalizability of the method (Reviewers dsg8, BZ3w).

— Computational cost analysis: We have included a discussion on the computational complexity of our method, comparing it to the previous state-of-the-art, PriViLege, in terms of parameters, training throughput, and inference throughput (Reviewer 3t7d).

— Ablation study expansion: We revised the ablation study to include standard deviations across multiple runs and highlighted the cumulative impact of removing multiple components, showing a significant performance drop when both controlled fine-tuning and consistency-guided incremental tuning are removed (Reviewer dsg8, BZ3w).

— Comparison with LoRA variants: We added experiments comparing our method with the latest LoRA variants, qLoRA and LoRA+ (Reviewer BZ3w).

---

### Decision · Action_Editor_bY8c · 2025-03-28

**Recommendation:** Accept as is

**Comment:**

The paper presents a novel and well-motivated approach to FSCIL with good experimental backing. The method effectively addresses forgetting and knowledge integration issues, making it a valuable contribution to the field. The clarity of writing, solid theoretical motivation, and empirical validation support acceptance.

**Audience:**

The paper is well-suited for researchers in few-shot learning, incremental learning, and foundation model adaptation, as well as life-long learning in general.

**Claims And Evidence:**

The paper proposes a Consistency-Guided Asynchronous Contrastive Tuning (CoACT) method for Few-Shot Class-Incremental Learning (FSCIL). The key claims include (1) CoACT improves knowledge retention by maintaining consistency across incremental learning stages.
(2) Asynchronous contrastive tuning mitigates forgetting and enhances representation learning. (3)  Empirical results demonstrate superior performance over state-of-the-art FSCIL methods on benchmark datasets.
This paper is reviewed by three experts. After rebuttal, all the reviewers are happy with the acceptance of this paper.  To sum up,  the paper provides substantial empirical evidence through rigorous experimental evaluations and ablation studies. The method’s effectiveness is validated across multiple benchmarks, with performance gains over existing approaches.

---

> ### Author Response · Authors · 2025-03-31
>
> Dear Action Editor and Reviewers,
>
> We are thankful for the great news. Your constructive feedback and comments have been invaluable in strengthening our work.